# Fabrication and Characterization of a Flexible Thin-Film-Based Array of Microelectrodes for Corneal Electrical Stimulation

**DOI:** 10.3390/mi14111999

**Published:** 2023-10-27

**Authors:** Natiely Hernández-Sebastián, Víctor Manuel Carpio-Verdín, Fabián Ambriz-Vargas, Francisco Morales-Morales, Alfredo Benítez-Lara, Mario Humberto Buenrostro-Jáuregui, Erik Bojorges-Valdez, Bernardino Barrientos-García

**Affiliations:** 1Centro de Investigaciones en Óptica, A.C. Loma del Bosque 115, León 37150, Mexico; victorcv@cio.mx (V.M.C.-V.); fambriz@cio.mx (F.A.-V.); fcomm@cio.mx (F.M.-M.); bb@cio.mx (B.B.-G.); 2CONAHCYT, Centro de Investigaciones en Óptica, A.C. Loma del Bosque 115, León 37150, Mexico; alfredbl@cio.mx; 3Laboratorio de Neurociencias, Departamento de Psicología, Universidad Iberoamericana Ciudad de México, Ciudad de México 01219, Mexico; mario.buenrostro@ibero.mx; 4Departamento de Estudios en Ingeniería para la Innovación, Universidad Iberoamericana Ciudad de México, Ciudad de México 01219, Mexico; erik.bojorges@ibero.mx

**Keywords:** microelectromechanical system, flexible electronics, sectioned surface electrode, selective electrical stimulation

## Abstract

The electric stimulation (ES) of the cornea is a novel therapeutic approach to the treatment of degenerative visual diseases. Currently, ES is delivered by placing a mono-element electrode on the surface of the cornea that uniformly stimulates the eye along the electrode site. It has been reported that a certain degree of correlation exists between the location of the stimulated retinal area and the position of the electrode. Therefore, in this study, we present the development of a sectioned surface electrode for selective electric stimulation of the human cornea. The proposed device consists of 16 independent microelectrodes, a reference electrode, and 18 contact pads. The microelectrodes have a size of 200 µm × 200 µm, are arranged in a 4 × 4 matrix, and cover a total stimulation area of 16 mm^2^. The proposed fabrication process, based on surface micromachining technology and flexible electronics, uses only three materials: polyimide, aluminum, and titanium, which allow us to obtain a simplified, ergonomic, and reproducible fabrication process. The fabricated prototype was validated to laboratory level by electrical and electrochemical tests, showing a relatively high electrical conductivity and average impedance from 712 kΩ to 1.4 MΩ at the clinically relevant frequency range (from 11 Hz to 30 Hz). Additionally, the biocompatibility of the electrode prototype was demonstrated by performing in vivo tests and by analyzing the polyimide films using Fourier transform infrared spectroscopy (FTIR). The resulting electrode prototype is robust, mechanically flexible, and biocompatible, with a high potential to be used for selective ES of the cornea.

## 1. Introduction

The electrical stimulation (ES) is a non-pharmacological therapeutic treatment in which a low-intensity electric current is delivered to a targeted tissue in order to activate, regenerate, and preserve its functions when it has been altered by trauma, congenital defects, diseases, or aging [1,2,3,4]. For instance, ES has been applied clinically for the management of pain, the healing of wounds and injuries to the spinal cord, and the treatment of degeneration of the visual function and diverse neuropathies [4,5,6].

The first recorded application of ES for the treatment of eye disorders and diseases was by Henri Dor in 1873, who used this approach to show the beneficial effects on patients with pathologies such as glaucoma, amblyopia, amaurosis, and optic atrophy. Later, in 1929 [7], Otfrid Foerster documented neural responsivity to ES of the visual cortex; similarly, in 1974, William Dobelle demonstrated that ES of the occipital cortex produces sensations of light (phosphenes) [8]. Until 2004, ES in ophthalmology aroused renewed interest since Chow et al., reported amelioration of residual vision in patients carrying an inactive sub-retinal prosthesis that produced only sub-threshold currents even in zones distant from the implant, which suggested that ES has a generalized neurotrophic effect on the retina [9]. This finding promoted investigations of the therapeutic potential of ES for treating visual disorders, such as age-related macular degeneration (AMD) and retinitis pigmentosa (RP) [1,4,6,10,11,12,13,14,15].

At present, eight sites along the visual path have been identified as being eligible for applying ES when improvement of the visual perception is desired [9]. Among these, we can mention the corneal surface (transcorneal ES) and the palpebral skin (transpalpebral ES), which are non-invasive approaches to activate the retina and downstream structures and then may exert therapeutic effects in patients. Both approaches have demonstrated protection effects on the retinal ganglion cells (RGCs) and on the photoreceptors preserved from degeneration processes. This protection mechanism may be due to the increase in certain bioactive factors, for example, insulin-like growth factor 1 (IGF-1), brain-derived neurotrophic factor (BDNF), and ciliary neurotrophic factor (CNTF) [6,13,14,15,16,17,18,19].

For the application of minimally invasive ES, two types of electrodes are in current use: the Dawson, Trick, and Litzkow (DTL) electrode, which comprises a thin metal-coated fiber, and the electrode for electroretinography (ERG-jet), which consists of a contact lens incorporating a gold-plated peripheral circumference. Both types of electrodes are placed on the surface of the cornea and can deliver an electric current to the eye uniformly along the electrode site. The two electrodes differ in the targeted area of the stimulated retina. The DTL electrode activates the visual association cortex, which processes input signals from the upper visual field. The ERG-jet electrode produces the activation of the anterior primary visual cortex, corresponding to the peripheral visual field. The ERG-jet electrode is also able to elicit brighter phosphene perception than the DTL electrode [2,20].

Considering that the stimulation area for these two types of electrodes corresponds to their mounting area and that they comprise only one stimulation element, the spatial resolution and selective stimulation are restricted. Unlike this, a multi-electrode array may allow us to select the target zone in a focused manner, as each element may be excited individually, which results in selective ES. Similarly, the type of stimulating electric signal can be selected as well, and special types of stimulation signals can be generated. The selective ES may be useful when treating loss of central vision, peripheral vision, or total vision, for example.

In this work, we report the design, fabrication, and characterization of a new sectioned surface electrode that, unlike conventional electrodes, enables the application of selective ES to the anterior surface of the cornea. The proposed design integrates an array of 16 independent microelectrodes, a reference electrode, and 18 electrode pads on a flexible ergonomic substrate. The fabrication process, based on surface micromachining and flexible electronics technology, uses only three materials: polyimide (PI), aluminum (Al), and titanium (Ti). In contrast to other works that integrate titanium/platinum or titanium/gold microelectrodes on polyimide substrates, we propose the use of an Al/Ti metallic bilayer as the structural material. In this way, it is possible to reduce the fabrication cost and obtain a biocompatible, efficient, and mechanically flexible device. In addition, the incorporation of materials widely used in the microelectronics industry allowed us to develop a simplified, reproducible, and low-temperature fabrication process. The fabricated prototype was electrically and electrochemically characterized; it exhibited a relatively high electrical conductivity and an average impedance comparable to other state-of-the-art electrodes (1.4 MΩ to 712 kΩ at a frequency range from 11 to 30 Hz). Finally, the biocompatibility of the electrode was evaluated by performing in vivo tests and FTIR spectroscopy.

The obtained results indicate that the prototype is non-cytotoxic and biocompatible with biological tissue. Therefore, the presented electrode is promising for applications of selective ES.

## 2. Materials and Methods

### 2.1. Design

The design of the proposed sectioned surface electrode is shown in Figure 1. The design consists of two levels of masks; the first level defines the metal structures composed of 16 independent microelectrodes, one reference electrode, 18 contact pads, and 18 conductor paths (see Figure 1a); these structures are positioned on a flexible ergonomic substrate made of polyimide. The 16 microelectrodes are arranged in a 4 × 4 matrix, distributed in an area of 4 mm × 4 mm (which is equivalent to the central region of the cornea). Each microelectrode has a size of 200 µm × 200 µm, and the stimulation site is 180 µm × 180 µm (see Figure 1c). The reference electrode, located along the periphery of the stimulation area, has a diameter of 4.5 mm and is much larger than the microelectrode size (see Figure 1b). As regards the contact pads, they were designed according to the commercial cable ZIF PS-2562 (Parlex, Shanghai, China). The total length of the electrode is 41 mm, and the diameter in the upper part (petals) is 12 mm (which is similar to the average diameter of a corneal contact lens).

The microelectrodes and contact pads are electrically connected by conductor paths. These metallic lines are approximately 36 mm long and 30 µm wide. The metal structures were designed considering a metallic bilayer composed of Aluminum/Titanium (Al/Ti) with thicknesses of 400 nm and 100 nm, respectively. The use of these two materials serves three purposes: to increase the electric conductivity, to achieve biocompatibility, and to reduce the fabrication cost.

The shape of the device and contact vias are defined by the second level of the mask; see Figure 1a (pink pattern). As it is shown, the design comprises all of the metallic structures, with the surface of the contact pads and that of the microelectrodes exposed for electrical connections to the external equipment and cornea, respectively. In addition, the shape of the device includes three petals for the stimulation area to facilitate its attachment to a contact lens. It is important to mention that the microelectrode array was designed to cover the central area of the cornea, but it can be easily adapted to cover either the peripheral or the total area.

### 2.2. Fabrication Process

The sectioned surface electrode for selective ES of the human cornea was fabricated using microelectromechanical system (MEMS) technology and flexible electronics. Figure 2 depicts the overall fabrication process. As shown, the process starts by considering an 18 µm-thick polymer layer as the flexible substrate. Polyimide 2611 (PI-2611, HD MicroSystems™, Parlin, NJ, USA) was chosen as the substrate material because of its excellent biocompatibility, flexibility, and film thickness selection. The PI-2611 was spun at 2000 rpm and cured at 380 °C for 2 h. To achieve the thickness of the substrate (18 µm), this process was repeated two times consecutively. Then, the PI-2611 surface was treated with oxygen plasma for 30 s using the RIE PE-100 equipment (Plasma Etch, Carson City, NV, USA). This treatment increases the surface roughness of the polyimide by around 10 nm, which enhances the adhesion between the polymer substrate and metals.

Subsequently, using a cathodic sputtering process, an Aluminum/Titanium metallic bi-layer of a thickness of 500 nm was deposited. The superimposed deposit of the metals and their parallel interconnection allow us to reduce the electrical resistance of the structures and maintain the biocompatibility of the device. Next, a layer of S-1813 photoresist (Rohm and Haas Electronic Materials LLC, Marlborough MA, USA) of 1.2 µm thickness was spin-coated and baked at 110 °C for 2 min. Following, a standard photolithography process was performed, and the metallic bi-layer was selectively etched in a solution containing deionized water (DI), hydrofluoric acid (HF), and nitric acid (HNO_3_) in a volume ratio of 20:1:1. Then, the photoresist layer was dissolved in acetone for 10 min.

After photoresist removal, a passivation layer of Polyimide 2610 (PI-2610, HD MicroSystems™, Parlin, NJ, USA) of 1.5 µm of thickness was spin-coated and cured. Then the PI-2610 surface was treated by oxygen plasma for 30 s, and a 100 nm-thick Al layer was deposited by cathodic sputtering. A standard photolithography process was carried out on the resulting aluminum layer, and this layer was selectively etched using a solution of phosphoric acid (H_3_PO_4_), glacial acetic acid (water-free CH_3_OOH), and nitric acid (HNO_3_) in a volume ratio of 25:7:1 (Al-etch). Then, to define the final shape of the device and contact vias, the polyimide was selectively etched using oxygen plasma for 6 h. Finally, the electrode was carefully detached from the silicon wafer in DI water.

### 2.3. Testing Methods

#### 2.3.1. Electrical Characterization

The electrical conductivity of the fabricated electrode was evaluated by measuring the electrical resistance of each pad-microelectrode path. The measurements were performed using the Summit 12000 Probe Station (Cascade Microtech, Inc., Beaverton, OR, USA) and the Keithley 4200A Parameter Analyzer (Tecktronix, Beaverton, OR, USA). The electrode was placed in the probe station at room temperature, and probes were positioned on the contact pad and on the microelectrode. A parameter analyzer was used to supply a sweep voltage from −5 V to 5 V (with steps of 0.01 V) and to measure the value of the electrical resistance.

Using these conditions, three types of measurements were conducted: (i) at wafer level, (ii) after being removed from the wafer (the flexible device is placed on a flat surface), and (iii) when the flexible device is placed on a concave surface (corneal-like curvature). In this way, it was possible to evaluate the integrity of the structures during and after the fabrication process and under conditions similar to those of operating conditions.

#### 2.3.2. Mechanical Test

The mechanical flexibility of the microelectrode is demonstrated by simply bending the device manually. Firstly, the microelectrode array was positioned on a concave surface with a radius of curvature of 8.0 mm (this curvature value is similar to that of a standard cornea), and then a bending moment was applied to the electrode by a pair of tweezers; in this latter condition, the attained curvature was twice that of the cornea, i.e., a resulting curvature radius of 4 mm was realized. After repeating this procedure ten times, the structural integrity of the whole device was confirmed by using an optical microscope.

#### 2.3.3. Electrochemical Impedance

The electrical impedance of the fabricated microelectrodes was measured using a three-electrode system, as shown in Figure 3. The microelectrodes under test were connected via probes and immersed in physiological saline solution (0.9% NaCl) at room temperature. A platinum (Pt) wire was used as the counter electrode (CE), and a silver/silver chloride (Ag/AgCl) electrode (Ref) served as the external reference.

Measurements were taken over a frequency range from 1 Hz to 100 Hz; an impedance analyzer (HIOKI IM3570, HIOKI, Ueda, Japan) was employed to supply a sinusoidal AC peak-to-peak voltage of 10 mV and to measure the impedance of the microelectrodes.

In this test, the frequency sweep was selected within 1 Hz and 100 Hz. However, the impedance value in the range of 11 Hz to 30 Hz was used to evaluate the microelectrodes, as the typical signal frequency values to stimulate the retina are around these values.

To verify that the device indeed diffuses a supplied signal throughout its surroundings, the performance of the electrode was evaluated using a variation in the scheme presented in Figure 3. In this case, the microelectrode array was immersed in physiological saline solution (0.9% NaCl) at room temperature; one probe was positioned on the contact electrode pad, and a second one was close to the array (at 1 mm and 5 mm away). A function generator (Hewlett Packard 33120A, Keysight, Santa Rosa, CA, USA) was employed to supply a sinusoidal peak-to-peak voltage of 10 mV at 11.8 Hz. The supplied signal was recorded by the second probe using a Tektronix TDS2004B digital oscilloscope (Tektronix, Beaverton, OR, USA).

The recorded signal was found to be similar in shape, frequency, and amplitude to the supplied signal, mimicking the response of a healthy retina.

#### 2.3.4. Biocompatibility Test

Polyamic acid (PAA) was used as the precursor of the polyimides 2610 and 2611, which are used as substrate and coating material in the microelectrode array, respectively. PAA is dissolved in a toxic compound based on N-methyl-2-pyrididone (NMP); therefore, for in vivo applications, it must be ensured that the PI solid film contains the least amount of NMP in its molecular structure. To demonstrate that this conforms to biocompatibility requirements, the PI films were analyzed using FTIR. The absorption spectra were taken for the PI films, which were cured at 300 °C, 350 °C, and 380 °C using the Bruker Vector 22 spectrometer (PerkinElmer, Puebla, México) in the range from 600 cm^−1^ to 2000 cm^−1^.

In addition, to further demonstrate the biocompatibility of the electrode prototype in contact with the biological environment, the microelectrode array was implanted onto the skull of a rat for two weeks. During this time, the surrounding biological tissue was visually examined for signs of inflammation and infection. After the implantation, the extracted array was examined under a microscope to assess any possible damages; the electrical continuity of the structures was confirmed to ensure their functionality.

We conducted a surgical procedure that involved administering a prophylactic treatment to the animal by using meloxicam (0.5 mg/kg, s.c.); in addition, it was anesthetized by Tiletamine/Zolazepam (60 mg/kg, i.m.). Once the rat was fully anesthetized, it was placed on a stereotaxic apparatus, and its head area was shaved. Next, the top head was sterilized by applying a benzalkonium chloride antiseptic solution; subcutaneous lidocaine (20 mg/mL) was administered in the area. Using a scalpel, a 2 cm incision was made from the anterior to posterior skull parts to expose the top region of the skull. The skin was retracted using bulldog clamps. The tissue covering the skull was scraped, and the microelectrode array was allocated within marked positions. Using a variable-speed drill tool, three holes were made at each marked point using a 2 mm tip (length 44.5 mm), taking care not to completely penetrate the skull.

The electrode was placed, and all screws were secured with dental cement. After surgery, nonsteroidal anti-inflammatory medication (meloxicam 2 mg/kg, s.c.) was administered in combination with an antibiotic (enrofloxacin 5 mg/kg, p.o.) for the next 24 h. The rat received post-operative care overnight, with periodic health checks, wound inspection, and electrode integrity assessment. This surgical procedure is supported by our previous work reported in [21].

## 3. Results and Discussion

### 3.1. Fabricated Device

Figure 4 shows the sectioned surface electrode fabricated. Figure 4a illustrates the flexible electrode placed on a glass concave surface to simulate its placement along the curvature of the cornea; a magnified image of the microelectrode array is shown in Figure 4b. The metal patterns of the microelectrodes and conductor paths before and after their removal from the silicon wafer are shown in Figure 4c,d.

As it is shown, we obtained a spatially selective, mechanically flexible, and technologically feasible electrode. In addition to this, incorporating materials widely used in the microelectronics industry allowed us to develop a simplified, reproducible, low-cost, and low-temperature fabrication process.

### 3.2. Mechanical Behavior

The mechanical test of the fabricated electrode validates, in a simple way, the mechanical flexibility and robustness of the prototype. After repeating the bending of the electrode ten times, the structural integrity of the whole device was confirmed by observing the structures under an optical microscope and by measuring their electrical conductivity. However, considering that structural flexibility is one of the most important characteristics of the proposed electrode, further tests should be performed by a universal testing machine to gain insight into its mechanical behavior.

### 3.3. Electrical Characterization

In the proposed design, the conductor paths have four different lengths: 3.60 cm, 3.62 cm, 3.64 cm, and 3.66 cm, while the electrode pads (3.5 mm × 0.28 mm) and microelectrodes (200 µm × 200 µm) maintain the same size. Considering these dimensions, the electrical resistance for each pad-microelectrode was analytically calculated as: 84.15 Ω, 84.61 Ω, 85.17 Ω, and 85.72 Ω, respectively. The experimental electrical resistance of each pad-microelectrode was measured for the three types of proposed conditions: at wafer level, after being released from the wafer, and when placed at the concave surface. The results obtained can be seen in Figure 5.

As it can be seen, the value of the electrical resistance measured for each pad-microelectrode (at the wafer level) fits very well with the analytical results, showing a maximum difference of 2 Ω. This small difference can be attributed to the main stages of the fabrication process and to the measurement system. On the other hand, the electrical resistance values measured after being removed (released) showed an average increase of 1 Ω and 2.1 Ω for the structures when positioned on a flat surface and on a concave surface, respectively. This increase is due to the stress that the metal film experiences during its deformation.

Regardless of the type of measurement (on the wafer or after being removed), the electrical resistance of structures with the same dimensions differs by less than ±1 Ω, which demonstrates a reliable, reproducible fabrication process that may be suitable for batch fabrication. Furthermore, it is important to mention that the proposed metallic bilayer (Al/Ti) allows us to obtain a low value of electrical resistance as it is a parallel metallic structure.

### 3.4. Microelectrode Impedance

Figure 6 shows the electrical impedance behavior as a function of frequency. As shown, the average impedance of a microelectrode decreases in magnitude from 6.4 MΩ at 1 Hz to 213 kΩ at 100 Hz. The impedance data that were collected for all 16 microelectrodes showed similar trends, indicating the good reliability of the fabricated device.

For the clinically relevant frequency range (from 11 Hz to 30 Hz), impedance values are 1.4 MΩ to 712 kΩ. These results are comparable to those of other microelectrodes, which reported impedance values on the order of several hundreds of kΩ [22,23,24,25,26]. This indicates that the fabricated microelectrode array might be appropriate for ES.

Additionally, the results obtained for the electrode performance test demonstrate that the supplied signal (a sinusoidal wave of 10 mV and 11.8 Hz) propagates with relatively slight losses through the saline solution. The resulting electric signals recorded at 1 and 5 mm away from the microelectrode array showed amplitudes of 9.68 mV and 6.78 mV, respectively, both at a frequency of 11.8 Hz. These results were complemented by Lissajous curves, which show that the supply and recorded signal present a 1:1 frequency ratio and a 0° phase shift; this means that the supplied electric signal is not affected by any physical defect of the electrode, which may cause non-uniform trajectories of the electric current flowing between the electrodes immersed in the saline solution. As for the decrease in amplitude, it is attributed to the electrochemical contact impedance observed at the electrode-saline solution interface.

### 3.5. Biocompatibility Test

The FTIR absorption spectra of polyimides 2610 and 2611 cured at 300 °C, 350 °C, and 380 °C in the range from 600 cm^−1^ to 2000 cm^−1^ are shown in Figure 7. The spectra show the main characteristic absorption peaks of the imide groups at around 721 cm^−1^ (imide IV), 1103 cm^−1^ (imide III), 1375 cm^−1^ and 1540 cm^−1^ (imide II), and 1726 cm^−1^ (imide I), as well as the characteristic absorption band of the solvent N-methyl-2-pyrididone (NMP) at approximately 1698 cm^−1^. As shown, the intensity of the imide peaks increases gradually as temperature increases; meanwhile, the intensity of the peak corresponding to the NMP solvent decreases.

All these absorption signals clearly indicate the presence of imide groups and confirm the synthesis of the polyimide. However, for in vivo applications, the minimum amount of NMP in the polyimide films must be guaranteed; therefore, the thermal curing temperature that we used in the fabrication process of the sectioned surface electrode was 380 °C. At this temperature, we can observe that the NMP absorption band is practically null. This confirms the successful synthesis of the polyimide with the desired composition; that is, we have achieved a flexible solid film with biocompatible characteristics.

The results of the implant surgery of the electrode onto the skull of the rat and those of the biocompatibility test are shown in Figure 8. The microelectrode array remained implanted for two weeks; during this period, we monitored the animal’s temperature, food and water consumption, body weight, and motor activity. No alterations were observed in any of the recorded parameters, and there were no signs of inflammation, infection, or changes in the skin or rat’s skull. Behaviorally, the animal did not exhibit distress, pain, erratic behavior, or abnormal movements. At this prototype development stage, evidence suggests that the animal showed no immune or toxic reactions (see Figure 8b,c). The region of the array in contact with the skull or any tissue displayed no signs of damage, as depicted in Figure 8d. Finally, the functionality of the device was demonstrated by checking the electrical continuity of the structures. All these data indicate that the prototype is non-cytotoxic and biocompatible with biological tissue. However, the next step for human use is to conduct cytotoxicity tests in both rat and human tissue.

## 4. Conclusions

We report the design, fabrication, and characterization of a new sectioned surface electrode that will allow selective electrical stimulation to be applied to the corneal surface. The proposed design integrates an array of 16 independent microelectrodes, a reference electrode, and 18 electrode pads on a flexible ergonomic substrate. The microelectrodes have a size of 200 µm × 200 µm, are arranged in a 4 × 4 matrix, and cover a total stimulation area of 4 mm × 4 mm (similar to the central region of the cornea). The proposed design can be easily adapted to cover either the peripheral or the total region of the cornea, depending on the stimulation target zone. The electrode was successfully fabricated using MEMS and flexible electronics technology using only three materials: PI, Al, and Ti. In contrast with other works that integrate Ti/Pt or Ti/Au microelectrodes on PI substrates, we proposed the use of Al/Ti as the structural material of the electrode. In this way, we manage to reduce the fabrication cost and obtain a biocompatible, efficient, and mechanically flexible device. In addition to this, incorporating materials widely used in the microelectronics industry allowed us to develop a simplified, reproducible, and low-temperature fabrication process.

The mechanical flexibility of the fabricated electrode was validated by positioning the electrode array on a concave surface with an 8 mm radius of curvature (cornea-like curvature) and by bending the device manually. This test validated, in a simple way, the integrity and robustness of the prototype. Hence, further tests should be performed to gain insight into its mechanical behavior. The electrical and electromechanical characterization shows a relatively high electrical conductivity of the structures and an average impedance value of 712 kΩ to 1.4 MΩ at a frequency range of 11 Hz to 30 Hz. In addition, the test in saline solution demonstrated that the electrode is capable of transmitting a stimulation signal to the surrounding medium. Finally, the biocompatibility test indicated that the prototype is non-cytotoxic and biocompatible with biological tissue. In future work, we are to conduct cytotoxicity tests in both rats and human tissue.

A limitation of the proposed electrode has to do with the wiring for its energization, since this is a factor for discomfort for the patient. The next step is to integrate the microelectrode array into a wireless power system.

The current state of the prototype represents a positive advancement towards selective electrical stimulation of the cornea.

## Figures and Tables

**Figure 1 micromachines-14-01999-f001:**
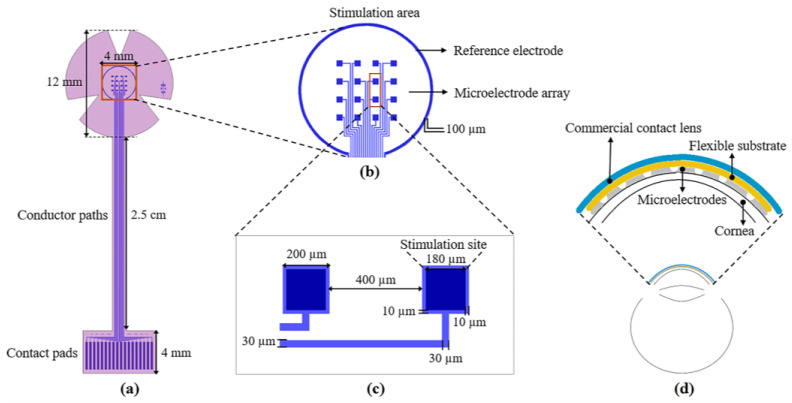
(**a**) Layout of the sectioned surface electrode for ES of the human cornea; it comprises of: 18 contact pads (bottom), 18 conductor paths (middle), 16 independent microelectrodes, and one reference electrode (top ring); (**b**) detailed view of the stimulation area; (**c**) key geometric parameters of the microelectrodes; and (**d**) diagram of the electrode mounted on the cornea; the inset shows the assembly of the electrode on the cornea.

**Figure 2 micromachines-14-01999-f002:**
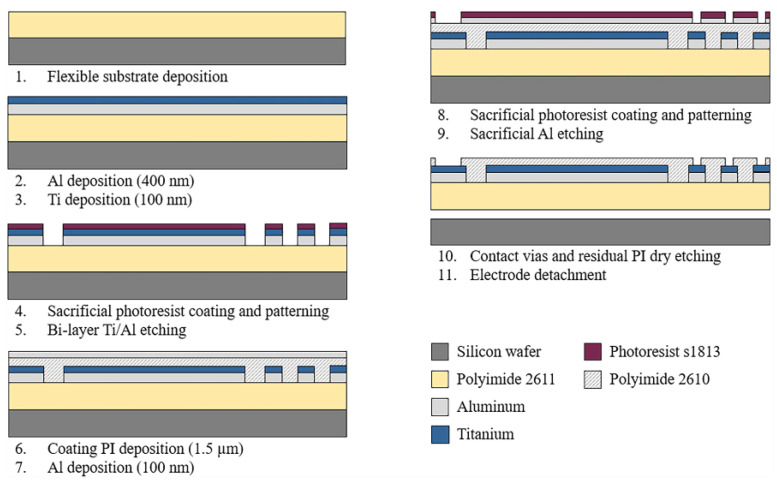
Schematic diagram of the fabrication process of the sectioned surface electrode.

**Figure 3 micromachines-14-01999-f003:**
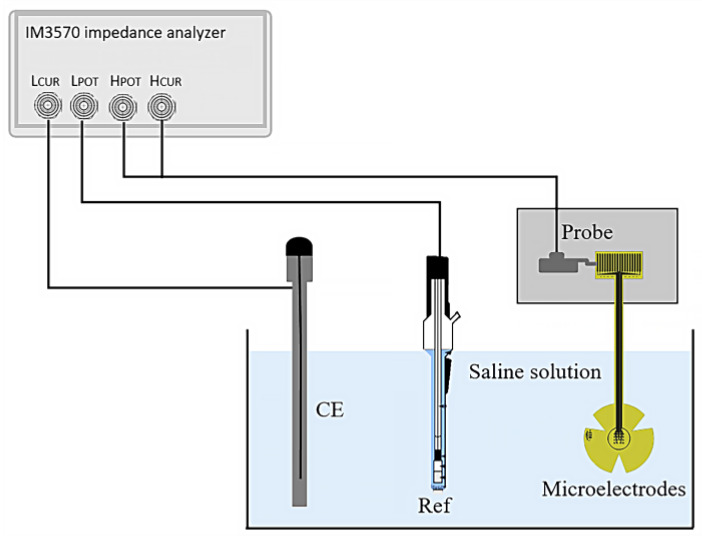
Set-up for the three-point impedance measurement.

**Figure 4 micromachines-14-01999-f004:**
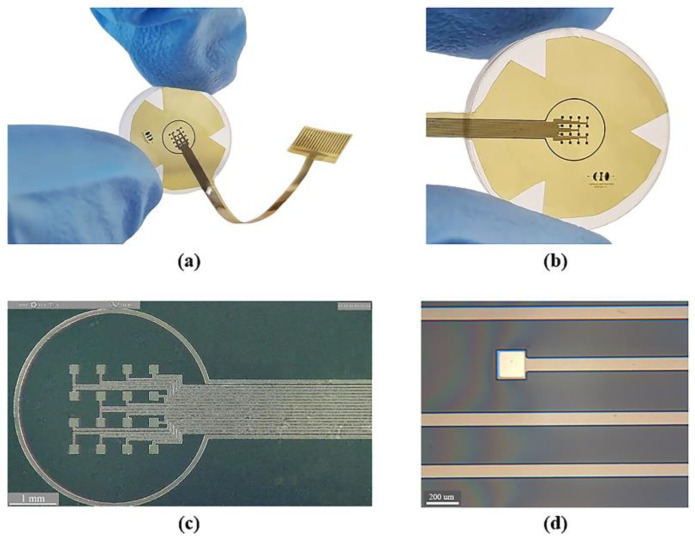
Fabrication process results: (**a**) flexible electrode positioned on a glass concave surface; (**b**) magnified view of the flexible microelectrode array; (**c**) microscopic view of the microelectrode array before release process (on a silicon wafer); and (**d**) microscopic zoom of one microelectrode and conductor paths after the release process.

**Figure 5 micromachines-14-01999-f005:**
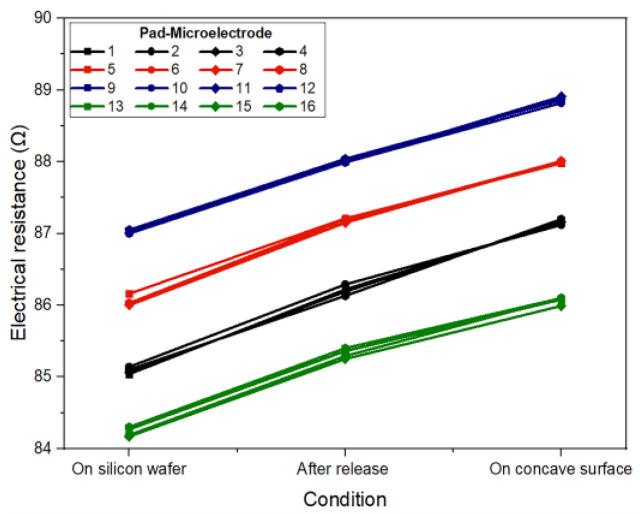
Experimental electrical resistance for each pad-microelectrode path considering three measurement conditions: on a silicon wafer, after being released from the wafer, and when the electrode is placed on a concave surface.

**Figure 6 micromachines-14-01999-f006:**
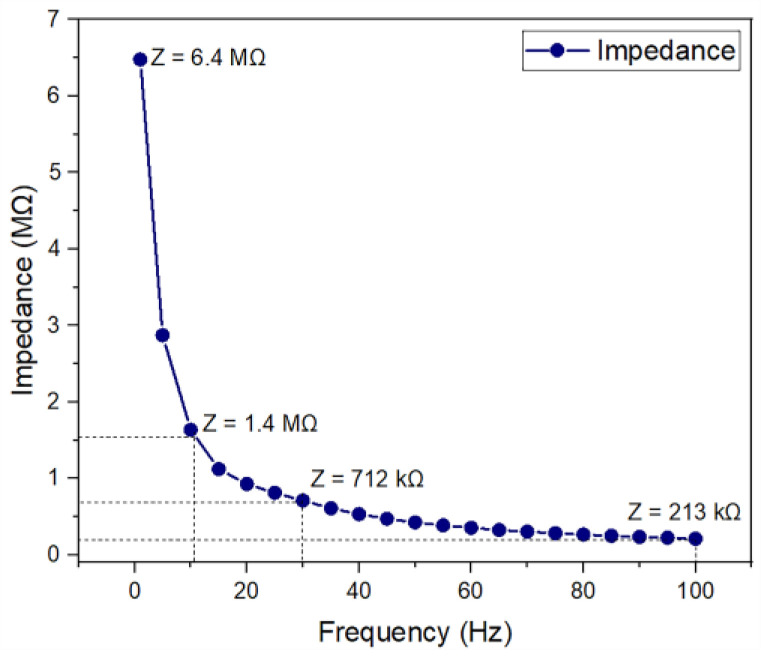
Electrical impedance measurement for the fabricated microelectrode.

**Figure 7 micromachines-14-01999-f007:**
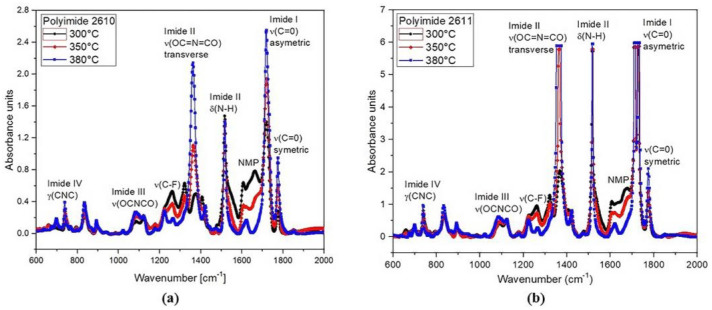
FTIR spectral absorbance in the region of 600 cm^−1^ to 2000 cm^−1^, at 300 °C, 350 °C, and 380 °C for (**a**) PI-2610 and (**b**) PI-2611.

**Figure 8 micromachines-14-01999-f008:**
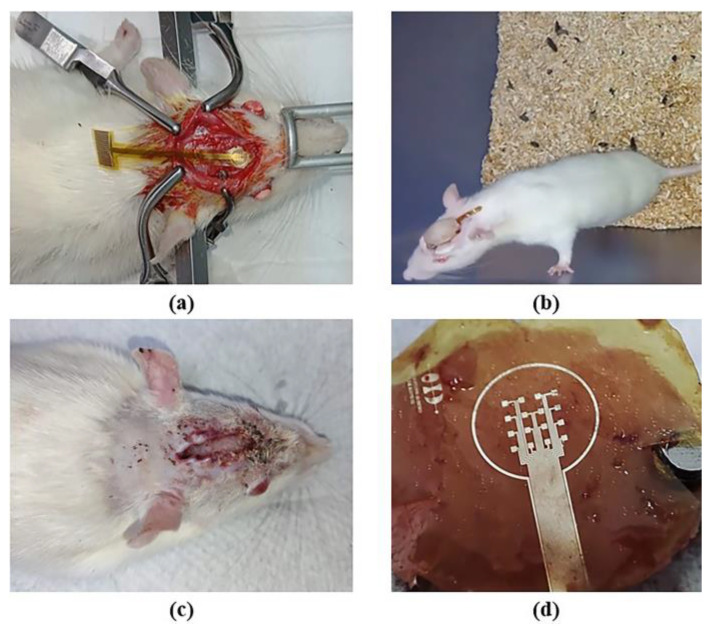
Implantation of a microelectrode array onto the skull of the rat: (**a**) surgical procedure; (**b**) implant evolution after one week; (**c**) implant removal after two weeks; and (**d**) extracted microelectrode array.

## Data Availability

The data that support the findings of this study are available from the corresponding author upon request.

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
