# Peer review of "Fabrication and Characterization of a Flexible Thin-Film-Based Array of Microelectrodes for Corneal Electrical Stimulation"

_micromachines, 2023, doi:10.3390/mi14111999_

Round 1

Reviewer 1 Report

In this manuscript, the authors proposed a sectioned surface electrode for selective  electric stimulation of the human cornea. Electrical, electrochemical and biocompatibility tests were used to verify the electrode performance. However, the existing test data were incomplete and the discussion was insufficient. The research direction and idea are good but not fully elaborated in the work. Thus, suggest major revision before further decision of acceptance or reject The detail comments are list as follows:

1. It is insufficient to prove the robust and mechanical reliability of the electrode only through electrical and electrochemical tests, and it is suggested to supplement the mechanical test data for further explanation.

2. Line 50, it is best to add recent research status in related fields.

3. Line 60-68, it is mentioned here that the advantage of this work compared with others is selective electrical stimulation, which is not elaborated in the following paragraphs. I suggest to add additional simulation data of global uniform electrical stimulation and local selective electrical stimulation to complete the discussion.

4. In Figure 1, the overall size of the electrode is missing (Figure 1(a)), and the relevant size is not mentioned in the manuscript; Figure 1 (c) is not an enlarged picture of the dotted line frame in Figure 1 (b). It is suggested to modify the position of the dotted line frame in Figure 1 (b). The leftmost 200 μm comment in Figure 1 (c) is incorrectly positioned. In addition, I suggest to add the assembly position diagram of the electrode and tissue in Figure 1.

5. In the electrical characterization section, the electrical resistance test for these three states can only prove that the impedance changes little during the electrode manufacturing process, but can not verify the use process, such as the process of implantation in the animal, and whether the resistance will change dramatically during the electrical stimulation process, so the proof of the reliability of the device is insufficient.

6. In Figure 6, change the vertical axis unit to MΩ, and then replace the vertical coordinate with Arabic numerals to make the picture more beautiful.

7. Line 267-276, it is not entirely convincing to judge biocompatibility simply by looking at whether biological tissue is intact.

Please correct the content with better expressions.

Reviewer 2 Report

This paper reported the design, fabrication and characterization of a new sectioned surface electrode. The fabrication process, based on MEMS and flexible electronics technology, was designed to optimizes the electrical properties of the electrode using an Al/Ti metallic bi-layer as a structural material.

1.      On page 2, line 69, the word “desing” should be “design”;

2.      On page 2, line 81, the word “Figura” should be “Figure”;

3.      Why were the materials Al/Ti used in the device instead of Ti/Au?

4.      On Lines 127 and 135, the HNO3 should use subscripts;

5.      In Figure 4, the scale bar should be given;

6.      On line 227, “crease of 1 Ω y 2.1 Ω”, what’s the meaning of ‘y’?

7.      The authors claim that the fabricated device could be used for electrical stimulation (ES), but the performance or behavior of the object after ES didn’t be shown.

N/A

Reviewer 3 Report

The authors developed an array of microelectrodes based on flexible thin-film for selective electric stimulation of the cornea. This device registered a high electrical conductivity at the frequency range from 11 Hz to 30 Hz. The electrical and electrochemical behavior of this device was validated with laboratory tests. In addition, the biocompatibility of the electrodes was reported by performing in-vivo tests. This manuscript is well-structured and written. This manuscript is well-organized and written. However, this manuscript can be improved by considering the following comments:
1.- The abstract should incorporate an introduction about the research problem.   2.- The introduction section should consider more discussions on the advantages and limitations of investigations about microelectrodes and their fabrication processes reported in the literature. Furthermore, this section should add the proposed device's main scientific contribution or innovation, considering the advantages and challenges compared to other technologies of microelectrodes for corneal electrical stimulation.   3.- The design subsection (2.1) can include more detailed information and better views of the proposed device's different elements, dimensions, and materials. For instance, information on the different layers (dimensions, thicknesses, and properties) of the layout used in the design of the microelectrodes (view Figure 1).   4.- The resolution of Figure 2 should be enhanced. For instance, the silicon wafer, aluminum, and titanium colors are similar.   5.- The authors can improve the description of the testing methods, including more information on environmental conditions (e.g., humidity and temperature) during the electrical characterization of the microelectrodes. How do these environmental parameters affect the electrical behavior of the microelectrodes?   6.- The authors should consider discussions about parameters that could affect the proposed device's performance. Discussions on the mechanical degradation of the microelectrodes array could be considered.   7.- The resolution of the response curves of Figure 5 can be enhanced.   8.- The authors could incorporate discussions of the limitations or challenges in the performance of the proposed device.   9.- The authors can consider future research on their proposed device.   10.- The format used in the references must be revised. For instance, the title of the journals must be abbreviated.   11.- The conclusions can be enhanced by considering the challenges and future research works of the proposed microelectrodes array.

The English grammar and style is good.

Round 2

Reviewer 1 Report

I suggest acceptance in this version, however, the signals are not shown or analyzed in this work.

The language is OK.

Reviewer 3 Report

The authors improved their manuscript based on the reviewer's comments.

English grammar is good.